# Efficacy and Safety of *Bifidobacterium longum* Supplementation in Infants: A Meta-Analysis of Randomized Controlled Trials

**DOI:** 10.3390/foods12244451

**Published:** 2023-12-12

**Authors:** Huangda Guo, Meng Fan, Tianjiao Hou, Yixin Li, Siyue Wang, Xueheng Wang, Hexiang Peng, Mengying Wang, Tao Wu, Yumei Zhang

**Affiliations:** 1Department of Epidemiology and Biostatistics, School of Public Health, Peking University, Beijing 100191, China; 2Key Laboratory of Epidemiology of Major Diseases (Peking University), Ministry of Education, Beijing 100191, China; 3Department of Nutrition and Food Hygiene, School of Public Health, Peking University, Beijing 100191, China

**Keywords:** *Bifidobacterium longum*, meta-analysis, randomized clinical trials, necrotizing enterocolitis, efficacy, safety

## Abstract

Background: Strategies to stabilize and support overall infant health by increasing the number of *Bifidobacterium longum* in the infant gut are of interest, but few studies have systematically addressed this issue. We aimed to evaluate the efficacy and safety of *Bifidobacterium longum* use in infants using meta-analysis. Methods: We searched PubMed, EMBASE, Cochrane Library of Systematic Reviews, and SinoMed for publications until 27 July 2022. The main outcomes of interest were weight gain, risk of necrotizing enterocolitis (NEC), and adverse events. Two authors independently performed study screening, risk of bias assessment, and data extraction. Outcome data were extracted from each included study and combined using mean difference (MD) or risk ratio (RR) and finally combined using a fixed-effect model or random-effect model. Results: A total of 4481 relevant studies were identified, of which 15 were found to be eligible for randomized controlled trials and were included in the meta-analysis. The combined extracted data showed that the intervention group containing *Bifidobacterium longum* had a significantly lower risk of NEC (RR = 0.539, 95% CI: 0.333, 0.874) compared to the control group. There was no statistical difference between the intervention and control groups regarding weight gain (MD = 0.029, 95% CI: −0.032, 0.090), the occurrence of adverse events (RR = 0.986, 95% CI: 0.843, 1.153), and serious adverse events (RR = 0.881, 95% CI: 0.493, 1.573). Conclusions: *Bifidobacterium longum* may significantly reduce the risk of NEC in infants as well as being safe; thus, further research evidence is needed on whether there is a benefit on weight gain.

## 1. Introduction

The initial colonization of the neonatal gut is a crucial and intricate process that significantly influences the development of the entire intestinal microbiota. This early microbial colonization not only shapes the complex ecosystem within the gut but also has the potential to impact various aspects of both intestinal and systemic health throughout an individual’s lifespan [1,2,3,4]. In recent years, the use and supplementation of probiotics have become prominent and dynamic areas of research, especially in the context of neonatal and infant health. Among the diverse array of probiotic species, specific members of the Bifidobacterium genus have gained considerable attention due to their potential pivotal roles in early human development [5].

*Bifidobacterium longum*, in particular, has gained significant attention as a noteworthy probiotic, making its way into a variety of functional foods and dietary supplements. A growing body of evidence suggests that *Bifidobacterium longum* may play a pivotal role in promoting optimal infant health. While the probiotic effects are believed to be strain-specific, earlier literature indicated that different strains of *Bifidobacterium longum* share common mechanisms to some extent [6]. Noteworthy correlations have been established, linking *Bifidobacterium longum* to favorable outcomes such as enhanced weight gain in infants [7] and a decreased incidence of the serious condition known as necrotizing enterocolitis (NEC) [8]. Furthermore, the metabolites produced by *Bifidobacterium longum* have been identified as crucial mediators in the maturation of the infant immune system [9] and as agents that instigate vital biochemical changes within the intestinal environment, effectively impeding the invasion and proliferation of potentially harmful pathogens [10]. Regarding the safety of *Bifidobacterium longum*, several studies have reported adverse effects in clinical trials, including gastrointestinal disorders [11,12,13], feeding intolerance [14], diarrhea [11,15], fever [12,13,15], respiratory symptoms [11,12,13,15,16], and even death [16,17,18]. However, due to the relatively limited sample size, it remains unclear whether these adverse effects can be conclusively attributed to *Bifidobacterium longum* supplementation.

In light of the increasing interest and a rising number of clinical trials investigating the efficacy and safety of probiotics in infants, it is crucial to emphasize that a thorough and comprehensive assessment of *Bifidobacterium longum* and its subspecies is still pending. The potential to improve overall infant health by stabilizing and boosting *Bifidobacterium longum* populations in the infant gut represents a highly promising and significant avenue. Nevertheless, it is noteworthy that a comprehensive evaluation of the aforementioned probiotic and its subspecies is yet to be conducted.

In this systematic review, our primary objective is to meticulously and comprehensively examine the current body of evidence regarding the efficacy and safety of *Bifidobacterium longum*. Our specific focus will be on its potential impact on infant growth, its role in preventing NEC, and a thorough evaluation of any potential adverse events associated with its supplementation. Through this comprehensive review, our aim is to illuminate the multifaceted aspects of *Bifidobacterium longum*’s influence on infant health, bridging existing knowledge gaps and establishing a solid foundation for future research in the critical domain of neonatal and infant care.

## 2. Materials and Methods

### 2.1. Literature Search

The review process was documented using Preferred Reporting Items for Systematic Reviews and Meta-Analysis guidelines (PRISMA). We conducted a thorough search across four electronic databases, specifically PubMed (https://pubmed.ncbi.nlm.nih.gov, accessed on 27 July 2022), EMBASE (https://www.embase.com, accessed on 27 July 2022), Cochrane Library of Systematic Reviews (https://www.cochranelibrary.com, accessed on 27 July 2022), and SinoMed (http://www.sinomed.ac.cn, accessed on 27 July 2022). This search encompassed studies published in both English and Chinese, covering the period from the inception of the databases up to 27 July 2022. In order to identify relevant trials, we employed a set of carefully chosen keywords, including infant, newborn, neonate, child, preschool, young, toddler, randomized controlled trial, controlled clinical trial, trial, and Bifidobacterium. For a more detailed breakdown of the search terms used, please refer to Appendix A.

### 2.2. Study Selection

Reviewers who underwent training were organized into pairs for the assessment process. Each pair consisted of two independent reviewers within the same group, and their task was to scrutinize the titles and abstracts of the studies to pinpoint those that potentially met the inclusion criteria. Studies that remained ambiguous based on titles or abstracts underwent a comprehensive examination through full-text reading. In case of discrepancies, each review group engaged in discussions to resolve inconsistencies. If disagreements persisted after the initial discussion between the two independent reviewers in a group, a third expert was brought in to conduct a detailed review and make a final determination. This meticulous and systematic approach ensured a thorough and reliable evaluation of the studies under consideration.

The “Participants-Intervention-Comparison-Outcomes-Study” (PICOS) approach was used for eligibility criteria. Only studies meeting the following criteria were included: (1) P (participants): the subjects were infants (including preterm infants) and young children aged 0–3 years without major diseases; (2) I (intervention): the treatment group consumed products containing *Bifidobacterium longum* or *Bifidobacterium subsp. longum* or *Bifidobacterium subsp. infantis* such as powder, milk, tablet, or capsule; (3) C (comparison): for comparisons, the current meta-analyses including both passive (e.g., milk without *Bifidobacterium longum* supplementation) and active (e.g., other probiotic supplementation) control conditions were eligible; (4) O (outcomes): the outcome of the studies included efficacy (growth and development indicators, gastrointestinal function and immune function) or safety (adverse events such as allergy, death, etc.); (5) S (study): randomized controlled trials (RCTs) were included, while quasi-RCTs were NOT eligible.

### 2.3. Data Extraction

Two proficient reviewers, namely H.G. and M.F., meticulously examined the encompassed studies and meticulously extracted a comprehensive set of data from each study. This extracted information encompasses the authors’ names, publication dates, countries conducting the studies, sample sizes, supplementation durations, baseline characteristics of the study population, as well as details regarding the probiotic species employed and their respective dosages. Also, the outcome parameters included: (1) weight gain at the end of the study, i.e., the increment in the last weight measurement during the study period compared to the measurement during the baseline period; (2) risk of necrotizing enterocolitis; (3) safety outcomes. To identify safety outcomes, we extracted pre-stated adverse events outcomes monitored by researchers or parents in each study. According to the primary study, any adverse events were assessed as serious adverse events if they were life threatening, caused permanent harm, resulted in hospitalization or prolongation of existing hospitalization, or resulted in persistent or significant disability or incapacity. The number of participants who experienced adverse events during the study period was extracted.

Continuous variables were evaluated or calculated with mean difference (MD) and 95% confidence intervals (CI) from the provided data. And categorical variables were evaluated or calculated with risk ratios (RRs) and 95% confidence intervals (CIs).

### 2.4. Quality Assessment and Statistical Analysis

We used the Cochrane Risk of Bias tool (ROB 2.0) to assess the quality of included studies [19]. Five main biases were evaluated: bias arising from the randomization process; bias due to deviations from intended interventions; bias due to missing outcome data; bias in the measurement of the outcome; and bias in selection of the reported results. There are three bias assessment criteria, “Low”, “High”, and “Unclear”, for each point. For a more detailed understanding of the specific items and criteria employed in the quality assessment using this tool, additional information can be found in the *Cochrane Handbook for Systematic Reviews of Interventions* [20]. Data analysis was performed using R, version 4.2.1 (package “meta”). We assessed heterogeneity among the studies by applying the *Q* test and inconsistency (*I*^2^) statistic. If *p* < 0.05 or *I*^2^ > 50%, we used the random-effects models to estimate the pooled estimates and corresponding 95% CI; otherwise, the fixed-effects models were utilized [21,22]. *p* values <0.05 suggested statistical significance.

## 3. Results

### 3.1. Study Selection and Characteristics

The flowchart delineating the literature search process is illustrated in Figure 1. Initially, a total of 4481 pertinent records were identified from diverse databases. Following the removal of duplicate articles and a meticulous review of titles and abstracts, 153 papers remained within the purview of consideration. Subsequently, after a comprehensive examination of the full-text articles, 138 records were excluded for the following reasons: non-compliance with the criteria for randomized controlled trials (30 records), involvement of non-healthy infants (39 records), absence of *Bifidobacterium longum* in the intervention group (32 records), examination of outcomes unrelated to our research interests (18 records), and incomplete data (19 records). Ultimately, 15 studies met the eligibility criteria for inclusion in the meta-analysis [7,8,11,12,13,14,15,16,17,18,23,24,25,26,27], and their respective data were extracted for further analysis.

Table 1 presents a comprehensive overview of the key characteristics of the eligible randomized controlled trials (RCTs). The geographical distribution of study locations was extensive, encompassing diverse countries across multiple continents. The trials were predominantly conducted in Asia, with a focus on countries such as India, Indonesia, and China. Additionally, research efforts extended to North America (specifically the United States), Oceania (notably Australia), and various European nations, including France, Italy, and Spain. In total, these RCTs involved 3152 participants, with participant numbers varying from 30 to 1099 per trial. The study population comprised a noteworthy proportion of female participants, ranging from 37% to 60%. The majority of participants fell into categories such as newborns, premature infants, and low birth weight babies. Furthermore, Table 1, along with Appendix A, furnishes detailed information regarding the outcomes of interest, mode of delivery, probiotic strains, dosage, antibiotic utilization, type of milk feeding, and the duration of supplementation for each RCT.

### 3.2. Study Quality

Following the Cochrane methodology, the risk of bias for the included studies was assessed, and the summary is presented in Figure 2A,B. Among the included trials, eleven were deemed to be at low risk of bias [8,11,12,13,14,15,16,17,25,26,27], primarily due to the availability of comprehensive information regarding the randomization process. In contrast, one trial [23] was classified as being at high risk of bias in the randomization process. The remaining trials had an unclear risk of bias in the randomization process [7,18,24]. Most of the studies demonstrated a low risk of deviations from the intended interventions [7,8,11,12,14,15,16,17,18,24,25,26,27], although two studies [13,23] were found to be at high risk in this regard. All trials were determined to have a low risk of bias related to missing outcome data. However, two trials [7,26] were considered to have an unclear risk of bias due to inconsistent measurements of the outcome. Eight studies were evaluated as having a low risk of bias concerning the selection of reported results [8,11,12,14,15,16,18,27]. Due to inadequate information regarding reporting and other potential sources of bias, most trials were categorized as having an unclear risk in these domains. In total, seven studies [8,11,12,14,15,16,27] were assessed as being at low risk, six studies [7,17,18,24,25,26] at unclear risk, and two studies [13,23] at high risk of bias. For a detailed breakdown of the risk of bias assessment for each included study, please refer to Appendix A.

### 3.3. Efficacy Outcomes

#### 3.3.1. Weight Gain

Our research sought to investigate the potential benefits of *Bifidobacterium longum* on weight gain in children. Upon analyzing the data extracted from the included studies, we identified five relevant studies [7,14,15,18,26] that presented data for the desired outcome. The pooled estimate for weight gain was 0.029 (95% CI: −0.032, 0.090), indicating a modest effect in favor of weight gain associated with *Bifidobacterium longum* supplementation. However, it is important to note that a high level of heterogeneity was observed among the studies (*I^2^* = 98%), as depicted in Figure 3.

#### 3.3.2. Risk of NEC

Six studies [8,17,18,24,25,27] investigated the impact of *Bifidobacterium longum* on the prevention of NEC. Among these, three studies [8,24,25] reported a significant risk reduction of NEC, while the other three studies [17,18,27] did not find statistically significant differences. It is worth noting that the point estimates from two non-significant studies [18,27] suggested an increased risk of NEC, but the smaller sample sizes led to wide 95% confidence intervals. In contrast, the three studies with larger sample sizes all demonstrated a significant reduction in the risk of NEC. To obtain a more comprehensive estimate of the effect, a pooled analysis was conducted, revealing that *Bifidobacterium longum* supplementation potentially led to a reduced risk of NEC during the follow-up period (relative risk = 0.539, 95% CI: 0.249, 0.874). Further details and a visual representation of these results can be found in Figure 4.

### 3.4. Safety Outcomes

In our safety analysis, we observed that the incidence of adverse events in the *Bifidobacterium longum* group did not appear to be higher than that in the control group (relative risk = 0.986, 95% CI: 0.843, 1.153), as depicted in Figure 5. Furthermore, there were no statistically significant differences in the occurrence of serious adverse events between the two groups (relative risk = 0.881, 95% CI: 0.493, 1.573), as illustrated in Figure 6. Notably, there was no heterogeneity detected in the analyses of serious adverse events, with a very low *I*^2^ value indicating consistency across the included studies. In particular, we conducted sensitivity analyses on studies involving only *Bifidobacterium longum* and its subspecies as an intervention group, with the results remaining stable (Appendix A). Additionally, considering variations in the length of gestation among studies, we stratified the included studies into those with gestation periods less than 37 weeks and those with gestation periods at or above 37 weeks for further analysis. Regarding adverse events, subgroup analyses based on gestation revealed that *Bifidobacterium longum* supplementation may be associated with a lower risk of adverse events in patients with a gestation period of less than 37 weeks (Appendix A, relative risk = 0.925, 95% CI: 0.859, 0.997), while the results for patients with a gestation period greater than or equal to 37 weeks were not statistically significant, consistent with the primary analysis (Appendix A). Concerning serious adverse events, we excluded the study with pregnancy periods of 37 weeks and above [16], and the results remained stable (Appendix A). Detailed information on adverse events and serious adverse events is provided in Appendix A.

## 4. Discussion

In our present study, we have undertaken a comprehensive review of the effectiveness and safety profile of *Bifidobacterium longum* in the context of infant health. Our analysis encompassed a total of 15 randomized controlled trials (RCTs) that adhered to the predefined inclusion criteria, enrolling a combined cohort of 3152 subjects. Our findings have demonstrated that the utilization of *Bifidobacterium longum* within the intervention group was associated with a notably reduced occurrence of necrotizing enterocolitis, in stark contrast to the control group. However, no statistically significant disparities were observed between the intervention group employing *Bifidobacterium longum* and the control group with regard to parameters such as weight gain and the prevalence of adverse events.

Previous research has advanced the notion that *Bifidobacterium longum* possesses the remarkable capacity to fully metabolize human milk oligosaccharides (HMOs) present in breast milk. HMOs, constituting the third most abundant solid component in human milk, following lactose and lipids, are impervious to digestion by human enzymes [28]. This metabolic prowess signifies an evolutionary adaptation of *Bifidobacterium longum* to thrive within the infant intestinal tract [29], whereby its metabolites wield direct influence over the host. Emerging data indicate that metabolites originating from *Bifidobacterium longum* can profoundly shape the maturation of the immune system [9]. Furthermore, notable reductions in the markers of chronic intestinal inflammation have been observed among preterm infants receiving *Bifidobacterium longum* subspecies [30]. These observations may offer valuable insights into the prophylactic effects of *Bifidobacterium longum* against NEC and the diminished occurrence of adverse events, as evidenced in our current meta-analysis. The metabolism of HMOs by *Bifidobacterium longum* culminates in the production of organic acids, predominantly lactic and acetic acids. These acids, in turn, lower the pH within the intestinal milieu. The acidification of the intestinal environment exerts inhibitory effects on the proliferation of pathogenic bacteria [10]. Thus, the capacity of *Bifidobacterium longum* to metabolize HMOs not only contributes positively to immune system development but also mitigates the risk of diseases commonly associated with preterm birth and intestinal inflammation, such as NEC. It is noteworthy to highlight the pivotal role of the feeding regimen in this context. Indeed, nutritional support for preterm infants has evolved over the past decade, including the earlier introduction of breastfeeding [31].

Our choice of weight gain as the other primary outcome variable is rooted in the paramount importance of infant growth and development, particularly in preterm infants. These neonates are highly susceptible to suboptimal postnatal growth due to various factors such as substantial insensible fluid losses, gastrointestinal tract immaturity, and heightened metabolic demands [32]. Ongoing research actively explores the role of intestinal microbiota composition in influencing weight gain [26]. The mechanisms through which prebiotic/probiotic combinations can potentially enhance growth encompass several facets. These include the improvement of feeding tolerance, the breakdown of indigestible carbohydrates by probiotic organisms (as illustrated by the HMOs mentioned earlier), and the production of essential vitamins and short-chain fatty acids by probiotic organisms [33,34]. In our present study, we did not observe any statistically significant disparities between the intervention and control groups regarding weight gain. This outcome may be attributed, at least in part, to the limitations stemming from our sample size. Given the pronounced heterogeneity observed in the data, we employed a random effects model to derive pooled estimates. Despite the lack of statistical significance in weight gain differences between the groups, it is crucial to acknowledge the multifaceted nature of infant growth and the potential impact of prebiotic/probiotic combinations on various aspects of development, as suggested by the existing literature. Future studies with larger sample sizes and refined methodologies are warranted to provide a more comprehensive understanding of the complex interplay between intestinal microbiota and infant growth. This heterogeneity may emanate from variations in gestational age, postnatal age, baseline weight, and the duration of follow-up within the study populations. Notably, three of the five studies exhibited relatively shorter follow-up periods and smaller sample sizes [14,18,26]. Conversely, the studies conducted by Agus et al. [15] and Al-Hosni et al. [7] featured relatively large sample sizes and consistently long-term follow-up, lending added credibility to their findings. Nonetheless, the point estimates greater than zero suggest a potentially positive influence of *Bifidobacterium longum* on weight gain despite the lack of statistical significance. In our safety analysis, it was observed that the intervention group receiving *Bifidobacterium longum* did not exhibit a different incidence of adverse events of any cause compared to the control group. Moreover, it is noteworthy that there was no statistically significant difference in the occurrence of serious adverse events between these two groups. However, the point estimates of safety analyses suggested a potential for supplementation of *Bifidobacterium longum* to decrease adverse events risk, where there is a plausible mechanistic link between *Bifidobacterium longum* and a reduced incidence of adverse events, likely attributable to improved immune function.

This meta-analysis presents several limitations. First, it is crucial to highlight that the studies incorporated into our safety analysis were also characterized by relatively short follow-up periods. This temporal limitation may preclude a comprehensive assessment of the long-term safety profile of *Bifidobacterium longum*. Consequently, the findings should be interpreted in the context of this temporal constraint. Second, due to the restricted number of endpoints focused on in the original literature included in this study, some endpoints that are considered important for infants, such as hospital-acquired infections, long-term growth neurodevelopment, all-cause mortality, and time to full feeds, were not included. Third, the overall quality of the included studies varied, with some studies having an unclear risk of bias and incomplete reporting of outcomes. These factors may introduce bias and affect the reliability of the findings, underscoring the necessity for further research to determine the exact effectiveness and safety of *Bifidobacterium longum* in infants. Fourth, since previous studies focused less on single *Bifidobacterium longum* strains, we also included studies with a mixture as an intervention group. Although we performed sensitivity analyses to verify the stability of the results, caution is still warranted regarding the interpretation of the findings. Fifth, it is important to consider the possibility of publication bias when interpreting our results. Notably, some researchers have suggested that probiotic trials are less likely to be published than antibiotic trials [35]. This highlights the need for the reporting and publication of high-quality scientifically valuable study results, whether positive or negative. Sixth, our study included both preterm and term infants, and possible differences in their risk profile, gut maturity, and response to probiotic supplementation make our findings need to be interpreted with more caution, though we used subgroup analyses in an attempt to address this problem. Finally, since the original literature did not provide information on the stage of NEC, we used overall NEC occurrence as an outcome metric, and future studies that explore the effect of *Bifidobacterium longum* on the risk of NEC incidence at different stages should be more in depth.

While the present systematic review and meta-analysis offer valuable insights into the efficacy and safety of *Bifidobacterium longum* in infants, several areas merit further exploration in future studies. First, to reinforce the current findings and gain a more comprehensive understanding of the long-term effects and safety of *Bifidobacterium longum* supplementation in infants, there is a need for more robustly designed randomized controlled trials with larger sample sizes and extended follow-up periods. Moreover, additional research efforts should be directed toward investigating the potential mechanisms by which *Bifidobacterium longum* exerts its effects. This could involve a closer examination of its impact on gut microbiota composition and function, immune system development, and metabolic processes. A deeper grasp of these underlying biological mechanisms can offer valuable insights into potential therapeutic targets. Lastly, it is crucial for future studies to evaluate the cost-effectiveness and feasibility of integrating *Bifidobacterium longum* supplementation into healthcare practices across diverse settings. This broader perspective will inform decisions regarding the practicality and economic viability of implementing such interventions in various healthcare contexts.

## 5. Conclusions

In conclusion, the systematic review and meta-analysis suggest that supplementing infants with *Bifidobacterium longum* may reduce the risk of necrotizing enterocolitis. Nevertheless, there was insufficient evidence to support its additional benefits in terms of promoting weight gain. It is crucial to acknowledge the variability in the quality of the studies included in this analysis, emphasizing the need for caution when interpreting the results. Further research is unquestionably needed to validate these findings, delve into the underlying mechanisms driving these effects, and assess the long-term consequences and safety of *Bifidobacterium longum* supplementation in infants. These forthcoming investigations will contribute to a more comprehensive understanding of the potential advantages and risks associated with this intervention.

## Figures and Tables

**Figure 1 foods-12-04451-f001:**
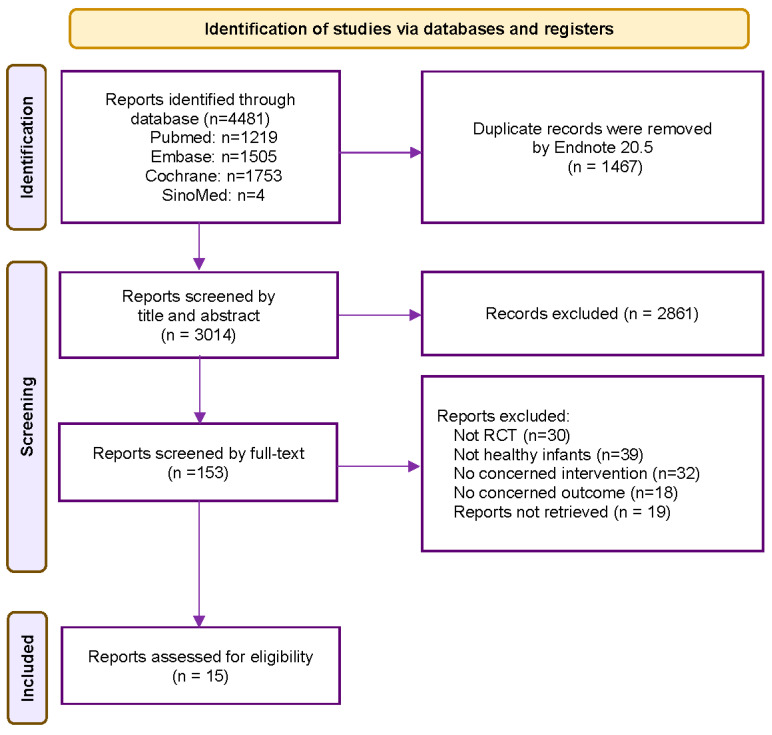
Flow chart depicting the study selection. After applying the inclusion and exclusion criteria, a total of 15 trials were included in the meta-analysis.

**Figure 2 foods-12-04451-f002:**
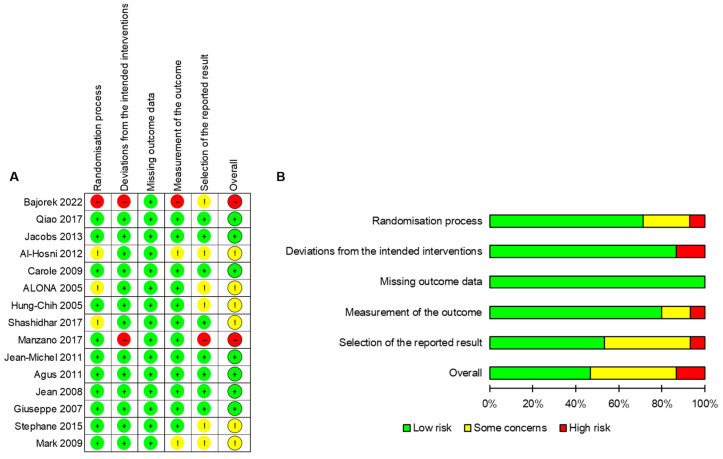
Study quality assessment [7,8,11,12,13,14,15,16,17,18,23,24,25,26,27]. The quality assessment of each included study is summarized in (**A**) risk of bias summary or is presented as a percentage across all included studies in (**B**) risk of bias graph.

**Figure 3 foods-12-04451-f003:**
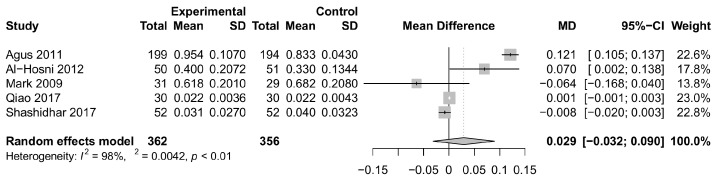
Forest plot illustrating the single study and summary weight gain afforded by *Bifidobacterium longum* [7,14,15,18,26]. CI, confidence interval; MD, mean difference; SD, standard deviation.

**Figure 4 foods-12-04451-f004:**
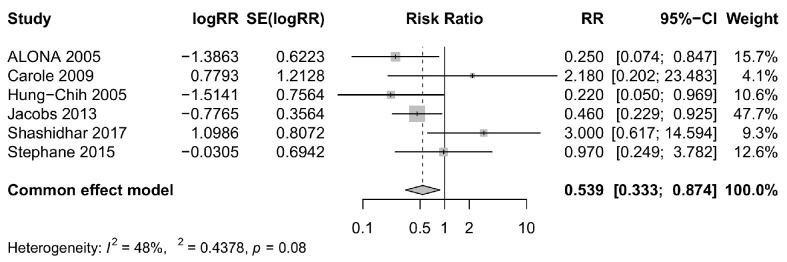
Forest plot comparing the rates of necrotizing enterocolitis for the intervention group versus control group [8,17,18,24,25,27]. CI, confidence interval; RR, risk ratio.

**Figure 5 foods-12-04451-f005:**
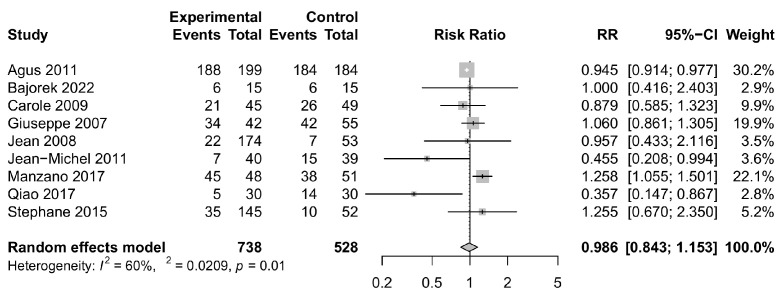
Forest plot comparing the rates of any adverse events for intervention group versus control group [11,12,13,14,15,16,17,23,27]. CI, confidence interval; RR, risk ratio.

**Figure 6 foods-12-04451-f006:**
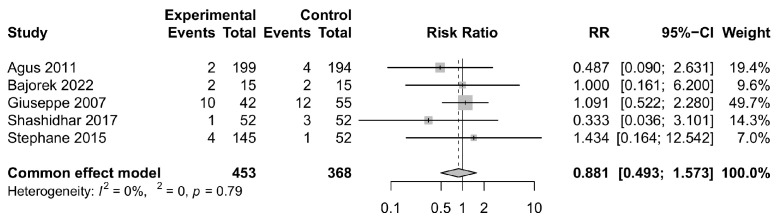
Forest plot comparing the rates of serious adverse events for intervention group versus control group [15,16,17,18,23]. CI, confidence interval; RR, risk ratio.

**Table 1 foods-12-04451-t001:** Basic characteristics of the studies included in the meta-analysis.

Study	Year	Country	Patients ^1^	Female	Birth Weight (g)	Gestational Age (Week)	Cesarean ^1^	Postnatal Age at Starting (Month)	Probiotic Strains	Total Dose (CFU/d) ^2^	Duration of Supplementation (Week)	Antibiotic Use	Type of Milk Feeding	Outcomes ^3^
Al-Hosni et al. [7]	2012	America	50/51	49.50%	778	26	22/30	4	*Bifidobacterium longum infantis, Lactobacillus rhamnosus GG*	5 × 10^8^	34	no	breast milk	a
Mark et al. [26]	2006	America	31/29	36.70%	1428	30	23/23	7	*Bifidobacterium longum*, *Bifidobacterium longum infantis Lactobacillus acidophilus, Bifidobacterium bifidum*	5 × 10^8^	5	no	-	a
Shashidhar et al. [18]	2017	India	52/52	54.80%	1223	31	27/38	0	*Bifidobacterium longum Lactobacillus acidophilus, Lactobacillus rhamnosus, Saccharomyces boulardii*	1.25 × 10^9^	4	no	breast milk	a, b, d
Qiao et al. [14]	2017	China	30/30	55.00%	1593	32	-/-	0	*Bifidobacterium longum, Lactobacillus acidophilus* *and Enterococcus faecalis*	1 × 10^7^	2	yes	breast milk	a, c
Agus et al. [15]	2011	Indonesia	199/194	48.30%	-	-	-/-	12	*Bifidobacterium longum BL999, Lactobacillus rhamonosus LPR*	3 × 10^7^	64	no	breast milk	a, c, d
ALONA et al. [24]	2005	America	72/73	44.10%	1131	30	56/57	0	*Bifidobacteria longum infantis, Streptococcus thermophilus, Bifidobacteria bifidus*	1.05 × 10^9^	6	yes	breast milk	b
Hung-Chih et al. [25]	2005	China	180/187	49.90%	1087	28	104/100	0	*Bifidobacterium longum infantis, Lactobacillus acidophilus*	-		no	breast milk	b
Jacobs et al. [8]	2013	Australia	548/551	52.00%	1055	28	359/377	0	*Bifidobacterium longum infantis BB02, Streptococcus thermophilus (TH–4), Bifidobacterium lactis (BB-12)*	1 × 10^9^		no	breast milk	b
Carole et al. [27]	2009	France	45/49	42.60%	1085	28	28/35	0	*Bifidobacterium longum BB536, Lactobacillus rhamnosus GG*	1 × 10^8^	2	yes	breast milk, preterm formula	b, c
Bajorek et al. [23]	2021	America	15/15	60.00%	1468	31	13/10	0	*Bifidobacterium longum infantis EVC001*	8 × 10^9^	4	yes	breast milk	c, d
Jean et al. [11]	2008	France	174/53	51.10%	3400	40	49/19	0	*Bifidobacterium longum BL999, Lactobacillus paracasei ST11, Lactobacillus rhamnosus LPR*	1.29 × 10^8^	16	no	-	c
Jean-Michel et al. [12]	2011	France	40/39	48.10%	3300	39	3/3	0	*Bifidobacterium longum BL999*	2 × 10^7^	16	no	preterm formula	c
Manzano et al. [13]	2017	Spain	48/51	51.50%	-	full-term (≥37 weeks)	8/9	6	*Bifidobacterium longum infantis R0033*	3 × 10^9^	8	no	breast milk, preterm formula	c
Giuseppe et al. [16]	2007	Italy	42/55	53.60%	-	39	15/24	0	*Bifidobacterium longum BL999*	2 × 10^7^	16	no	preterm formula	c, d
Stephane et al. [17]	2015	France	145/52	48.70%	1170	29	115/39	0	*Bifidobacterium longum, Bifidobacterium lactis*	1 × 10^9^	3	yes	breast milk, preterm formula	b, c, d

^1^ The left side of the slash is the intervention group, and the right side is the control group. ^2^ The dose given was the full dose of all the supplemented strains. ^3^ a, weight gain; b, necrotizing enterocolitis; c, any adverse events; d, serious adverse events.

## Data Availability

Data are contained within the article and Appendix A.

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
