# Peer review of "Efficacy and Safety of Bifidobacterium longum Supplementation in Infants: A Meta-Analysis of Randomized Controlled Trials"

_foods, 2023, doi:10.3390/foods12244451_

Round 1
Reviewer 1 Report
Comments and Suggestions for Authors
Dear Authors,
The use of probiotics in newborns, especially preterm infants is still controversial hence the demonstration of the efficacy and safety of Bifidobacterium longum is very valuable. The meta-analysis was based on 15 studies in which as many as 1,671 newborns, mostly preterm infants, received the probiotic.
The study was well designed, conducted and comprehensively described with one exception. I consider that due to the different age of inclusion in the study (0-12 months), length of probiotic use (2-64 weeks), and gestational age (26-?), the evaluation of the effect of Bifidobacterium longum on weight gain is not authoritative and should not be carried out. Admittedly, in the limitation study, the authors raise the issue of a large variety of patients, but still, based on this meta-analysis, it is impossible to conclude whether and how the probiotic affects weight gain and the conclusion that it probably does not is an overinterpretation.
Please improve the writing of Bifidobacterium longum- like all species (and strains) it should be in italics.
Despite the above comment, I believe that the work is a significant contribution to the development of probiotic therapy in neonatology.
Author Response
Please see the attachment. Thank you for your comments!

Reviewer 2 Report
Comments and Suggestions for Authors
The paper by Huangda Guo and colleagues deals with a metanalysis focused on the relation between Bifidobacterium longum and both weight gain and risk of necrotizing enterocolitis in infants. The analyses are well conducted and the statistical approach used is, in the opinion of this reviewer, well-engineered.
The only improvement that can be made is relative to the introduction in the main text of a simplified version of the supplementary table S4 relative to adverse events. In order to give a frame for the type of adverse events, at least cite a purged list in the main text.
No grammar issue was detected.
Author Response

(The authors gave the same response as above.)

Reviewer 3 Report
Comments and Suggestions for Authors
Comments to Authors:
1. Authors should justify how different their meta-analysis than “Batta, V.K., Rao, S.C. & Patole, S.K. Bifidobacterium infantis as a probiotic in preterm infants: a systematic review and meta-analysis. Pediatr Res (2023). https://doi.org/10.1038/s41390-023-02716-w”
2. Is the meta-analysis conducted by authors is according to Preferred Reporting Items for Systematic reviews and Meta-analyses (PRISMA)?
3. Line 25: ratio instead of raito.
4. Line 77-79: provide we side links.
5. Line 93: Provide full-form of PICOS.
6. Line 97: why Bifidobacterium subsp. Infantis?
7. Figure 1 provide is according to PRISMA flow diagram of study selection?
8. Table 1: specify other strains as well.
9. In table 1: the dose given was for only Bifido? Please provide clarity.
Comments on the Quality of English LanguageMinor improvements and typo
Author Response

(The authors gave the same response as above.)

Round 2
Reviewer 3 Report
Comments and Suggestions for Authors
Authors revised the manuscript extensively, thus I hereby recommend to accept this revised version.